# Apple-Net: A Model Based on Improved YOLOv5 to Detect the Apple Leaf Diseases

**DOI:** 10.3390/plants12010169

**Published:** 2022-12-30

**Authors:** Ruilin Zhu, Hongyan Zou, Zhenye Li, Ruitao Ni

**Affiliations:** Department of Mechanical, Nanjing Forestry University, Nanjing 210037, China

**Keywords:** apple leaf disease, Apple-Net, CA, FEM, YOLOv5

## Abstract

Effective identification of apple leaf diseases can reduce pesticide spraying and improve apple fruit yield, which is significant to agriculture. However, the existing apple leaf disease detection models lack consideration of disease diversity and accuracy, which hinders the application of intelligent agriculture in the apple industry. In this paper, we explore an accurate and robust detection model for apple leaf disease called Apple-Net, improving the conventional YOLOv5 network by adding the Feature Enhancement Module (FEM) and Coordinate Attention (CA) methods. The combination of the feature pyramid and pan in YOLOv5 can obtain richer semantic information and enhance the semantic information of low-level feature maps but lacks the output of multi-scale information. Thus, the FEM was adopted to improve the output of multi-scale information, and the CA was used to improve the detection efficiency. The experimental results show that Apple-Net achieves a higher mAP@0.5 (95.9%) and precision (93.1%) than four classic target detection models, thus proving that Apple-Net achieves more competitive results on apple leaf disease identification.

## 1. Introduction

Apples are rich in medicinal and nutritional value, providing material guarantees for human health. Meanwhile, the apple industry is one of the most widely planted fruit industries in the world [1]. However, due to apple leaf disease, apples will suffer serious quality deterioration and yield losses. If not treated in time, it will not only bring great harm to apple production but also cause panic among consumers, resulting in serious economic losses [2,3]. Therefore, it is very important to identify apple leaf diseases accurately.

The traditional identification methods of apple leaf diseases are mostly based on the long-term planting experience of fruit growers to judge and identify the diseases and then take corresponding measures such as spraying chemicals to prevent and control them. The problem with this method is that the time cost is high and the efficiency is low. In the process of manual diagnosis of diseases, the fruit growers will make false judgments and miss judgments due to the diversity of leaf diseases and the large number of leaves, leading to further worsening of apple diseases [4].

In recent years, the field of artificial intelligence in agriculture has been continuously developing. The detection and identification of apples and other fruits are of great significance for agricultural production [5]. With the rise of machine learning, researchers use different algorithms to establish plant disease diagnosis models. Qin et al. [6] proposed a region segmentation method based on K-means clustering by considering the classification algorithms of k-nearest neighbor, random forest, and support vector machine. They identified and detected four common alfalfa leaf diseases in apples, and the accuracy rate reached 80%. The research provides a feasible solution for the pathological image segmentation and image recognition of alfalfa leaf disease. Rothe et al. [7] used the method of extracting Hu matrix features to train the adaptive neurofuzzy inference system and finally achieved 85% accuracy. They provided a basis for the detection of apple leaf disease by identifying the leaf disease on cotton plants. Compared with the previous manual identification, these methods using machine learning algorithms to identify plant diseases not only reduce the time spent in identifying diseases but also improve the accuracy of disease identification. However, it is necessary to manually select appropriate features according to knowledge in relevant fields during training, and there are strict requirements for experimental conditions, which are low in universality and difficult to promote.

Compared with traditional machine learning, deep learning can select features by itself without human intervention, which solves the problem that selecting features requires relevant professional knowledge, and its performance can be continuously improved with the increase in data size [8,9]. A convolutional neural network (CNN) has been widely used in image recognition because of its excellent performance. It accomplishes the task by reducing the demand for image preprocessing and improving recognition accuracy [10]. With the continuous improvement and optimization of CNN models, they are widely used in agricultural production. Scholars have established plant disease diagnosis models based on CNNs [11,12]. H. Sun et al. [13] proposed a lightweight CNN model that can detect apple leaf disease in real-time. By using the MEAN block and the apple inception module, a new apple leaf disease detection model was established. The experimental results show that MEAN-SSD can achieve 83.12% mAP detection performance and 12.53 FPS (frames per second). Li et al. [14] proposed an apple leaf disease detection model based on an improved, faster RCNN. The experimental results showed that the average accuracy of the model for apple leaf disease under natural conditions reached 82.48%. YADAV et al. [15] proposed a novel, CNN-based model for the recognition and classification of apple leaf diseases. The model applies a contrast stretching-based preprocessing technique and fuzzy c-means (FCM) clustering algorithm for the identification of plant diseases. It achieved 98% accuracy in 400 image samples. The work of the above scholars has promoted the development of plant leaf detection to a certain extent. Even though the deep learning methods mentioned above have made a breakthrough in the application of apple leaf disease detection, there are still shortcomings in the existing research. These can be understood as having two aspects: first, the detection model has high accuracy, but the training data are small, and the generalization ability is poor; second, the generalization ability of the model is good, but the accuracy is not high. Therefore, it is urgent to find a better method that can consider both the accuracy and generalization of the model and be widely used in the identification of apple leaf diseases.

In this paper, we propose an improved YOLOv5 [16] model for apple leaf disease identification called Apple-Net to solve the problem that the existing models cannot consider the diversity and accuracy of diseases. We selected five diseases of apple leaves to build a leaf detection dataset, and these pictures were collected in both a static background and a wild environment to improve the diversity of identification. A novel apple leaf disease identification model called Apple-Net is proposed. This model introduces the Feature Enhancement Module (FEM) and Coordinate Attention (CA) into the YOLOv5 model [17,18], and moreover, we use the GAN algorithm [19] to reduce the noise of raindrops in the collected field leaf images. Apple-Net has the following advantages: Compared with other target detection models, it ensures a higher precision value and can detect various apple leaf diseases without being affected by the environment or noise. The rest of this paper is organized as follows: Section 2 describes the datasets and Apple-Net; the experimental results are described in Section 3; and discussions of comparative research are given in Section 4. Finally, this paper is concluded in Section 5.

## 2. Materials and Methods

In this section, the dataset of apple leaves used in this paper is first introduced. Secondly, the network structures of YOLOv5 and Apple-Net are introduced and compared. Then the method of preprocessing the collected real-time blade image is introduced, and finally the performance evaluation index of the experiment is introduced. Figure 1 is the step flow chart for this section.

### 2.1. Data

In the experiment, we selected the apple leaf pictures with a moderate size and clear image from Baidu PaddlePaddle’s image library and then used software to label and produce the dataset required for the experiment. We expanded the dataset by adopting some conventional image processing methods, such as filtering, translation, and rotation [20]. In this work, we mainly detect and identify five common diseases of apple leaves, so the dataset contains the following five kinds of images of apple leaf diseases: alternaria blotch, brown spot, gray spot, mosaic, and rust disease. The samples for each disease are listed in Figure 2.

In this work, our dataset contains 12,500 images. We selected 2000 pictures as the training set and 500 pictures as the test set for each disease. Then we labeled different diseases. Label numbers 0–4 represent alternaria blotch, brown spot, gray spot, mosaic, and rust diseases separately. The details of the dataset are shown in Table 1.

### 2.2. Training Model

The training model adopted by Apple-Net is based on YOLOv5. The structure of YOLOv5 is divided into four parts: input, backbone, neck, and prediction [21]. Considering the problems of different receptive fields of each feature map caused by multiple scales of apple leaves and the insufficient expression ability of network learning features, a FEM was adopted on the neck end feature pyramid of YOLOv5 [22], and a CA module was adopted after the CSP structure of the backbone to improve model detection efficiency [23]. The structure of the Apple-Net model is shown in Figure 3. Meanwhile, we put the original YOLOv5 model in Figure 4 and compared the similarities and differences between the two models.

#### 2.2.1. Part of Input

The input adopted a mosaic data enhancement method [24]. The dataset was enriched by splicing pictures through random scaling, random clipping, and random arrangement. In addition, it adopted the adaptive anchor box calculation method. The length and width of the anchor box was initially set for different training sets. In the process of training the network, the network outputted the prediction box after setting the anchor box, and then compared it with the real box [25]. The difference between the two was calculated through the built-in algorithm and finally, through the reverse update and iteration of the network parameters [26].

#### 2.2.2. Part of Backbone

The backbone is a significant part of YOLOv5s, where the CSP structure is set. This part is mainly composed of a focus structure and multiple CSP structures. The focus structure is used for slicing, and Figure 5 [27] shows the process of this structure. The original 608 × 608 × 3 image was input into the focus structure. By using the slicing operation, it first became a feature map of 304 × 304 × 12, and then, after a convolution operation of 32 convolution cores, it finally became a feature map of 304 × 304 × 32. This structure can make full use of the feature information in the feature map and avoid information loss.

The CSP structure was to divide the original input into two branches, perform convolution operations to halve the number of channels, then perform Bottleneck × N operations on one of the branches, and then connect the two branches. The structure diagram of CSP1_X is shown in Figure 6. The input was divided into two branches. One branch passed through the CBL first, then passed through multiple residual structures (Bottleneck × N), and then performed a convolution. The other branch performed a convolution directly. Then the two branches were connected, followed by BN, Leakey ReLU, and CBL [28]. The function of this structure is to make the model learn more features. 

A CA module was adopted in the part of YOLOv5 to reduce the amount of calculation for detecting the input image and improve the detection efficiency, which was set after the CSP structure of the backbone section. The structure of this module is shown in Figure 7.

In this part, the input characteristic map was divided into two parts: height and width, and then average pooling was carried out, respectively, to obtain the corresponding characteristic map. 

After that, the two characteristic graphs were connected and reduced through a c/r convolution layer. After the convoluted characteristic map passed through the BN (batch normalization) layer, the sigmoid activation function was used to obtain the characteristic map F. The sigmoid activation function [29] and its formula are as follows:(1)$$\delta \left(x\right)=\frac{1}{1+e^{-x}}$$

We then used a 1×1 convolution kernel to convolve the characteristic graph F in order to get the directional feature graphs, which have the same number of channels as the original map. Then, the weight of height and width was obtained from the directional feature image through a sigmoid activation function.

Eventually, we calculated the original feature map to obtain the feature map with attention weights on both components [30].

#### 2.2.3. Part of Neck

The neck part of YOLOv5 uses a combination of a feature pyramid and pan structure that we call the FPN + PAN structure. The advantage of using this structure is that it can obtain richer semantic information and enhance the semantic information of the low-level feature map. However, it lacks the output of multiscale information [31]. In order to solve this problem, we added a FEM in the neck section.

A FEM is mainly divided into two parts. One of the parts is called the multi-branch convolution layer. It enables the input characteristic map to obtain more receptive field information by adopting the method of extended convolution. In this part, there are three layers called the extend convolutional layer, the BN layer, and the ReLU activation layer [32]. They are similar in many ways, but the most distinctive aspect is the velocity of convolutional expansion. Specifically, they all use 3 × 3 convolutional kernels, and the expansion velocities D of the three levels are 1, 3, and 5, respectively. The extended convolution [33] can expand the receptive field exponentially without losing resolution. In the convolution operation of extended convolution, the elements of the convolution kernel are spaced, and the size of the space depends on the expansion rate, which is different from the adjacent elements of the convolution kernel in standard convolution operation. 

The other part is the average pool layer [34], which fuses the receptive field information in different structural layers of the first part and then feeds back the fused information to the network so as to improve the prediction accuracy of the model on multi-scale images. The branch pooling layer is used to fuse information from different parallel branches to avoid introducing additional parameters. In the training process, the average operation was used to balance the representation of different parallel branches so that a single branch could achieve reasoning in the testing process. The structure of this part is shown in Figure 8. The multi-branch convolution layer provided different-sized receptive fields for the input characteristic map through cavity convolution. Then each layer passed through the BN layer and ReLU activation layer, respectively, and finally, the average pooling layer was used to fuse the information from the three branch receptive fields to improve the accuracy of multi-scale prediction [35].

#### 2.2.4. Part of Prediction

In the prediction part, we still used the DIOU loss of YOLv5 as the loss function of the bounding box, and the loss function is as follows:(2)$$\mathrm{DIOU}\_\mathrm{Loss}\;=1-\left(IOU-\frac{Distance\_2^{2}}{Distance\_C^{2}}\right)$$
where *IOU* represents the intersection and union ratio of the target frame and the prediction frame [36].

### 2.3. Detection Model

We took the best model obtained in the training process as the detection model. Since there may be raindrops in the apple leaf images collected in the field environment, we preprocessed the apple leaf images collected in real time. We first judged whether there were raindrops in the input detection pictures, used GAN network to reduce the noise of the images, and then detect the disease on the images after noise reduction [37].

#### 2.3.1. Raindrop Image Recognition Network

The texture and color of apple leaves blurred when there were raindrops, and according to this, we judged whether there were raindrops on the leaves. Firstly, the photos of apple leaves taken on sunny days were used as a comparison library and processed with gray and binary. Then the SIFT (scale invariant feature transform) algorithm [38] was used to extract the feature points of the leaves, and the extracted key points were vectorized. After that, the extracted local features were used as the matching template. The vectorization matrix [39] of key points in the template diagram is as follows:(3)$$M_{ij}=\left[\begin{array}{ccc} m_{11} & \cdots & m_{1j}\\ \vdots & \ddots & \vdots \\ m_{i1} & \cdots & m_{ij} \end{array}\right]$$
then we input the image to be detected, carried out gray and binary processing on the detected image, used the SIFT algorithm to extract leaf surface feature points, vectorized the extracted key points, and took the extracted local features as the observation map. The vectorization matrix [40] of key points in the observation chart is as follows:(4)$$R_{ij}=\left[\begin{array}{ccc} r_{11} & \cdots & r_{1j}\\ \vdots & \ddots & \vdots \\ r_{i1} & \cdots & r_{ij} \end{array}\right]$$
then the template map and observation map were measured for similarity, and the measurement formula [41] is as follows:(5)$$d\left(M_{\mathrm{ij}},R_{\mathrm{ij}}\right)=\sqrt{{\left(m_{ij}-r_{ij}\right)}^{2}}$$
where $\;m_{ij\;}$ and $\;r_{ij}$ represent the eigenvalues of key points in the template matrix and observation matrix, respectively. We set the threshold value, determined whether there were raindrops in the picture according to whether d is greater than or less than the threshold, and performed raindrop noise reduction on the image.

#### 2.3.2. Noise Reduction

We used the attentive GAN algorithm [42] to reduce the raindrop noise of apple leaves. Attentive GAN is a raindrop removal method based on a single image. It uses the generation confrontation network, where the generation network generates an attention map through the attention recurrent network and generates a raindrop-free image through the contextual auto encoder together with the input image.

### 2.4. Experimental Environment and Evaluating Indicators

In this paper, the deep learning framework is Pytorch. The experimental configuration environment is as follows: CPU is Intel(R) Core(TM) i5-7300HQ with 16g of memory. GPU is NVIDIA GTX 1050, CUDA version 11.3 and Python 3.6 as the compiler language.

In the field of target detection, the commonly used performance evaluation indicators include precision value, accuracy rate, recall rate, mean average precision (mAP@0.5), and the frames per second (FPS). The main evaluation indicators we used in this experiment were recall rate, precision, and mAP@0.5. The formula for the recall [43] rate is as follows:(6)$$\;R=\frac{TP}{TP+FN}\times 100\%\;$$
where R represents the recall rate, *TP* is the number of positive categories predicted by the model as positive examples, and *FN* is the number of positive categories predicted by the model as negative examples, that is, the number of samples misclassified by the model. The formula for the precision [44] value is as follows:(7)$$P=\frac{TP}{TP+FP}\times 100\%\;$$
where P represents the precision value and *FP* is the number of positive samples compared with the originally negative examples predicted by the model. The mAP@0.5 was used to measure the algorithmic performance of the detection network. It is suitable for single-label and multi-label image classification and calculation. The formula for mAP@0.5 is as follows:(8)$$\mathrm{mAP}@0.5=\frac{\sum_{i=1}^{k}P\left(i\right)\Delta R}{N}$$
where *k* is the number of samples in the test set, P(i) is the size of the precision when *i* samples are recognized, ΔR(i) is the change of the recall rate when the detected samples change from *i* to *i* +1, and *N* is the number of categories in the multi-class detection task [45].

## 3. Results

### 3.1. Experimental Result

Apple-Net trained on 10,000 training pictures and 2500 testing pictures. The size of the input picture was 640×640, the batch size was set to 4, the number of training rounds was set to 100, the initial learning rate was set to 0.01, and the model updated every 10 rounds. We used the Adam optimization [46] algorithm and set the weight attenuation coefficient to 0.0005. After 100 rounds, the loss reached a stable and minimum value. Figure 9 shows the change in training losses. From the figure, we can see that the loss function eventually tended to zero, which proves that the Apple-Net model works well. Next, we can evaluate various performance indicators of the model.

Figure 10 shows the mAP@0.5, recall, and precision values of Apple-Net for training and testing datasets over 100 epochs. It can be found that the precision of Apple-Net exceeds 90% after 80 epochs. At the end of 90th epochs, the highest mAP@0.5 of 95.9% is achieved on the testing dataset.

### 3.2. Comparison of the Accuracy of Different Network Models

In this paper, we compared the four most advanced object detectors with our improved model to evaluate the performance of the Apple-Net model. Figure 11 shows the mAP@0.5 of these models. We can see that the algorithm models in the YOLO series have certain advantages in the detection of apple leaf diseases. The rise amplitude of the mAP@0.5 curve of YOLOv4 and YOLOv5 is relatively similar because YOLOv5 is slightly improved on the basis of YOLOv4, and there is not much improvement in accuracy. Furthermore, the mAP@0.5 curves of the SSD [47] and Faster RCNN [48] models rise more zigzag, which may be because they are used as two-phase target detection algorithms. They select too many candidate regions in the feature map, and the adjacent windows have an abundance of repeated information and many invalid regions. Therefore, the training accuracy rises slowly and is not high. Apple-Net adds a CA module, which can reduce the amount of calculation on the input image and improve detection efficiency. Meanwhile, the added FEM can enhance the output of multiscale information. Therefore, its mAP@0.5 curve can quickly reach the maximum value, and the effect is stronger than the YOLOv5 model before improvement.

The other performance indicators of these models are shown in Table 2. We can see that compared with the original YOLOv5 model, the accuracy rate of the Apple-Net model is 93.1%, an increase of 5.5%; the recall rate is 94.4%, an increase of 4.1%; and the mAP@0.5 value is 95.9%, an increase of 6.1%. Compared with the last YOLOv4 version of the YOLO series, the precision achieves an increase of 8.6%, and the other evaluation index has made a great leap. Compared with other target detection algorithms, the Apple-Net model also has excellent performance indicators.

### 3.3. Ablation Experiment

An attention mechanism was added to the Apple-Net model, which enabled the model to independently select the focus position of the image, enhance the expression ability of network learning features, and ultimately improve the training accuracy. To obtain better experimental results, we further conducted ablation experiments for the selection of the attention mechanism module. We add a CA module, SE (squeeze-and-excitation) module, ECA (efficient channel attention) module, and CBAM (convolutional block attention module) to the YOLOv5 model, respectively [49]. The performance indicators are shown in Table 3. Figure 12, Figure 13, Figure 14 and Figure 15 show the performance index curves of four different attention mechanisms.

As shown in Table 3 and Figure 16, each performance index of a SE and CBMA module is very close. The convergence speed of a SE loss function is slightly faster than that of a CBMA, and the convergence value is also similar because the structures of the two are very similar. A CBMA adds special attention on the basis of a SE, which can enhance the feature distribution weight of the object to be detected in space and channel dimensions and improve model accuracy, but it is also slower than a SE in terms of loss convergence speed. When they compress the input feature map, they will lead to an interdependence between channels, so the accuracy will be lower than other channels. Although, after 100 rounds of training, the performance indicators of an ECA are better than those of a SE and CBMA and close to those of a CA, its loss function converges more slowly, and the convergence value is also large. Because an ECA only performs dimensional processing on the feature map, it enhances the distribution weight of the detection object but does not eliminate the interference of useless features when fitting the results. During the training, the convergence speed of the improved YOLOv5 with a CA module was faster than that of other models, and the performance index was also better. This is because a CA can enhance the expression ability of mobile network learning features. It can transform and change any intermediate feature tensor in the network and output tensors of the same size. Moreover, it can encode channel relationships and long-term dependence through accurate location information, which is a capability that a SE and CBMA do not have. In addition, it can reduce the amount of calculation on the input image and accelerate convergence [50].

After training the model, we used the detection model to detect apple leaf disease. We first preprocessed the collected real-time image. The attentive GAN algorithm was adopted to reduce raindrop noise and enhance the image of the pictures. Eventually, all processed images were detected to obtain the final detection result. The inspection image of apple leaf disease is shown in Figure 17.

Figure 17 shows the detection results of the real-time apple leaf images collected. It can correctly detect leaf diseases collected in the wild, such as gray spot, alternaria blotch, and mosaic. This shows the robustness of Apple-Net, which can detect and identify different apple leaf diseases under various environmental conditions.

## 4. Discussion

Plant diseases are a significant threat to global apple production and supply, and the latest artificial intelligence technology needs to be applied to agriculture to control diseases. It is very important for apple growth and production to detect and treat disease in apple leaves. With the continuous development of artificial intelligence algorithms, a variety of new technologies have been applied to the detection of apple leaf disease. They can help farmers improve the efficiency of disease detection and then improve the quality of apple fruit. However, questions that remain unanswered include how to balance the accuracy and generalization of the model. For disease identification, the priority is to correctly identify the type of disease. In this case, the accuracy of the model is very important. Conversely, assuming that the model can correctly identify the type of disease, the model needs to be able to identify more disease types at this time, so the model needs to have excellent generalization ability. The proposed Apple-Net system is suitable for this task.

In this paper, we establish an apple leaf disease detection model that is based on YOLOv5. It has excellent performance indicators on the apple leaf dataset constructed in this paper and can meet the needs of a diversity of leaf diseases. Table 4 shows the comparison between Apple-Net and some existing apple leaf disease detection methods. Most of the current methods are used to detect five or fewer apple leaf diseases. The models proposed in reference [51] only identify two classes of apple leaf diseases and may not be able to cope with the diversity of apple leaf diseases. The model proposed in reference [44] was used to detect four apple leaf diseases; the accuracy rate does not exceed 90% when the training set is large. The accuracy of the models proposed in references [6,43,52] in detecting apple leaf diseases exceeded 90%, while the number of training sets was too small, which made it easy to produce overfitting and poor generalization ability. Conversely, while the number of datasets in references [14,53] is large and the accuracy rate of the proposed models is more than 97%, their values of mAP@0.5 do not reach 85%, which means that the detection effect of the model on each specific apple leaf disease is very poor. The accuracy of the models proposed in references [54,55] exceeded 95%, and the number of training sets was large, which was suitable for apple leaf diseases. However, the mAP@0.5 value was still low. While the references in Table 4 used different datasets, the apple leaf dataset constructed in this paper covers the leaf categories in the above literature and has a large amount of data that can adapt to the diversity of apple leaf diseases. In addition, the performance index of the model has obtained competitive results.

Therefore, Apple-Net can accurately detect apple leaf diseases, which is of great value to the application of artificial intelligence in agricultural production.

## 5. Conclusions

In this paper, to detect various apple leaf diseases, we selected clear apple leaf pictures with moderate size from Baidu PaddlePaddle’s image library and used Make Sense software to label and produce the dataset required for the experiment. After that, we proposed Apple-Net to detect apple leaf diseases. We introduced the FEM and CA methods into the YOLOv5 model structure to improve the model training effect. The experimental results showed that Apple-Net achieved excellent results across multiple indicators. The mAP@0.5 value of this model is 95.9%, which is 9.4%, 14.7%, 5.6%, and 6.1% higher than SSD, Faster RCNN, YOLOv4, and the original YOLOv5. On the precision indicator, Apple-Net achieved 93.1%, which is about 5.5% higher than YOLOv5. In the detection phase, we added a raindrop image recognition network to distinguish the pictures of apple leaves with raindrops and used the attentive GAN algorithm to enhance images and finally improve the accuracy of real-time detection. Therefore, Apple-Net can effectively detect the disease of apple leaves, which provides an effective method for improving the yield and quality of apples. As a small target disease detection model, Apple-Net can also be used for the disease identification of other agricultural products, such as oranges, pears, and other fruits.

However, there is still a shortcoming in this paper: our model does not take model parameters or lightweight into account, so it will take a long time to train the model. In future work, more research can be improved in the following aspects: (1) We plan to select a suitable lightweight model to reduce the amount of calculation and parameters of the model when the precision value is excellent. (2) We plan to deploy the improved Apple-Net model to mobile devices, making it easier for fruit growers to detect diseases.

## Figures and Tables

**Figure 1 plants-12-00169-f001:**
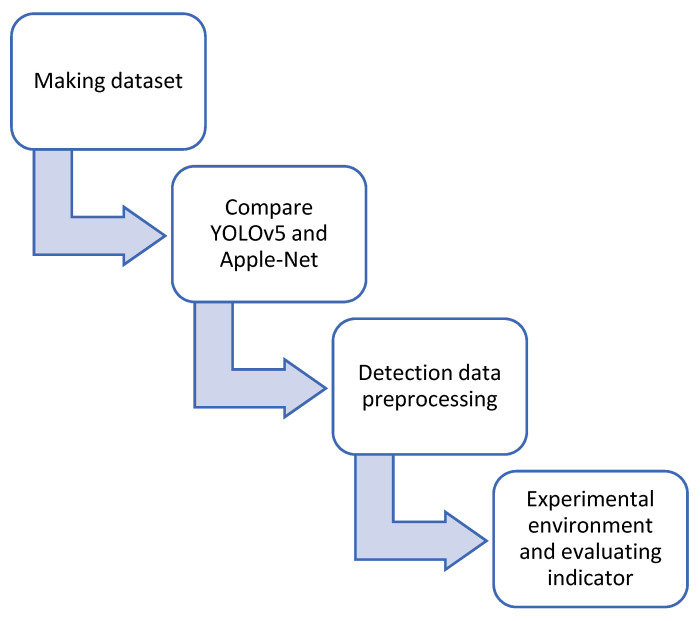
Step flow chart for our research process.

**Figure 2 plants-12-00169-f002:**
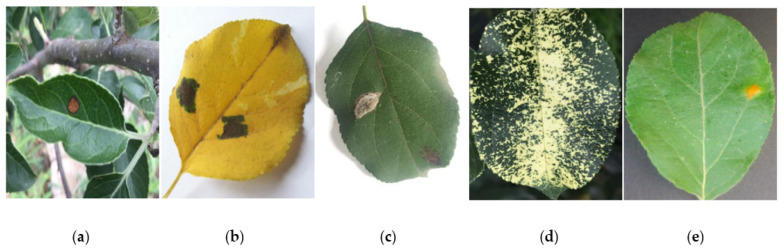
Five kinds of apple leaf diseases. (**a**) alternaria blotch, (**b**) brown spot, (**c**) gray spot, (**d**) mosaic, (**e**) rust.

**Figure 3 plants-12-00169-f003:**
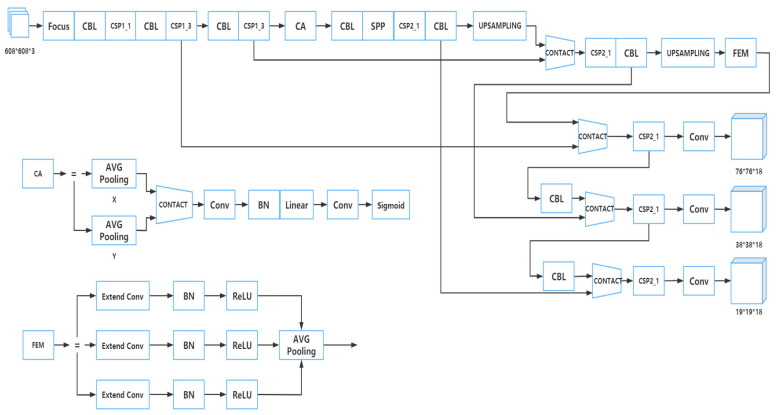
Network structure diagram of Apple-Net.

**Figure 4 plants-12-00169-f004:**
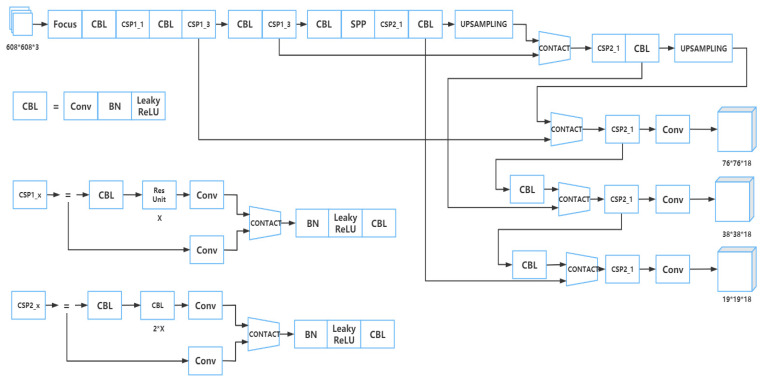
Network structure diagram of YOLOv5.

**Figure 5 plants-12-00169-f005:**
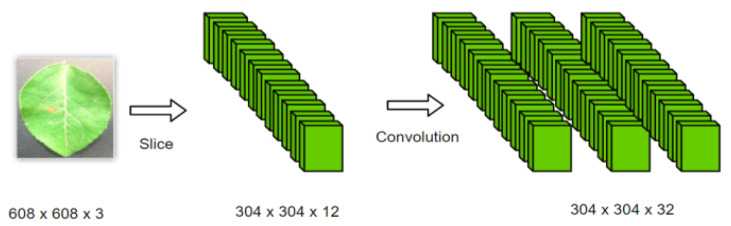
Slicing process of the Focus.

**Figure 6 plants-12-00169-f006:**
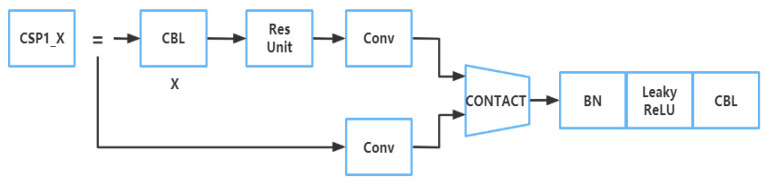
Structure diagram of CSP1_X.

**Figure 7 plants-12-00169-f007:**
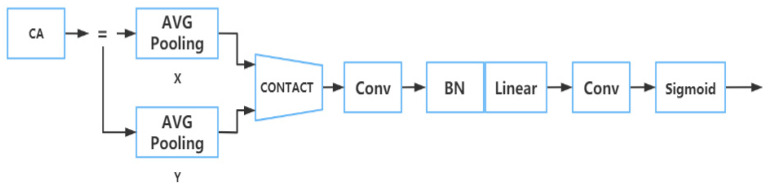
The structure diagram of CA module.

**Figure 8 plants-12-00169-f008:**
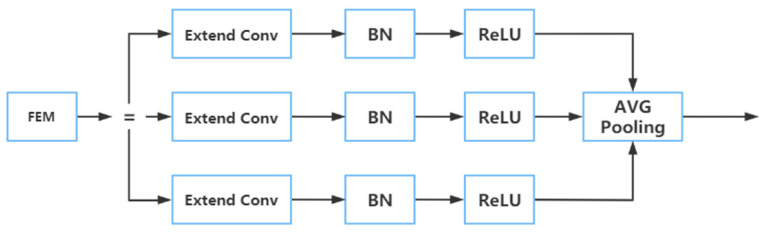
The structure diagram of FEM.

**Figure 9 plants-12-00169-f009:**
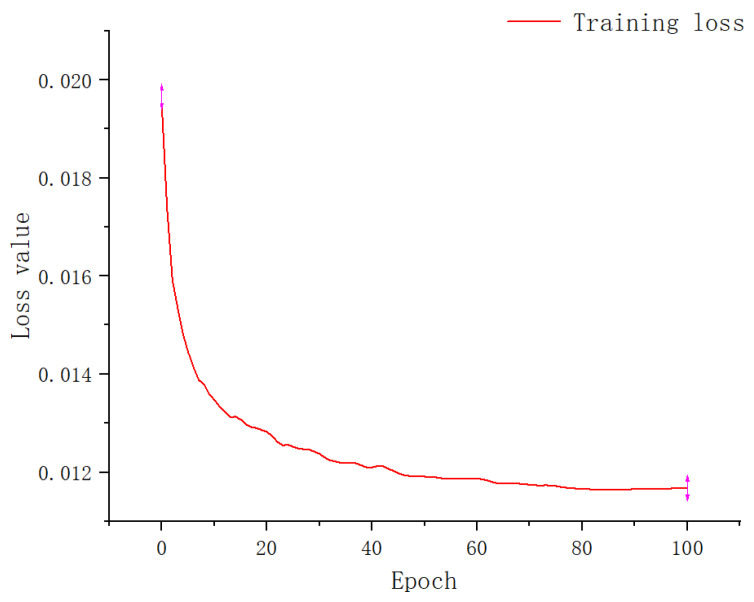
Training loss curve.

**Figure 10 plants-12-00169-f010:**
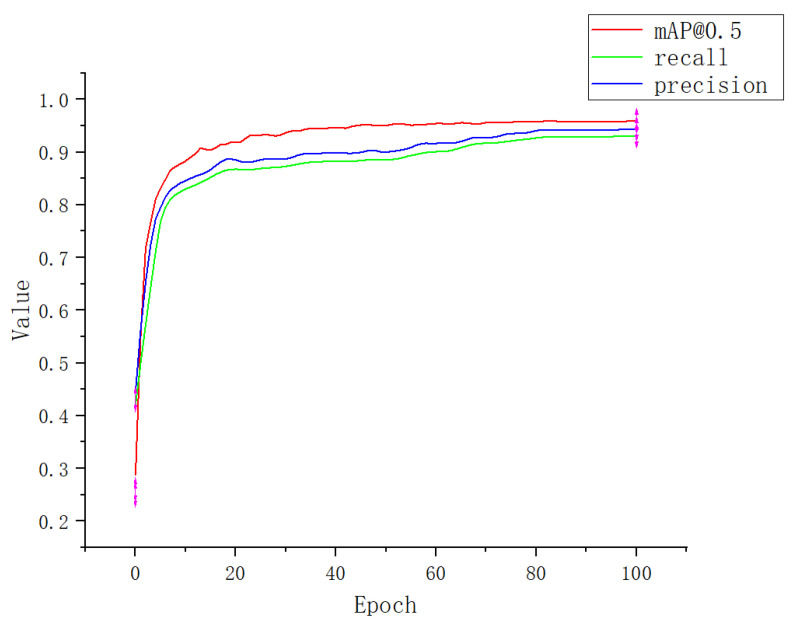
Performance index curve.

**Figure 11 plants-12-00169-f011:**
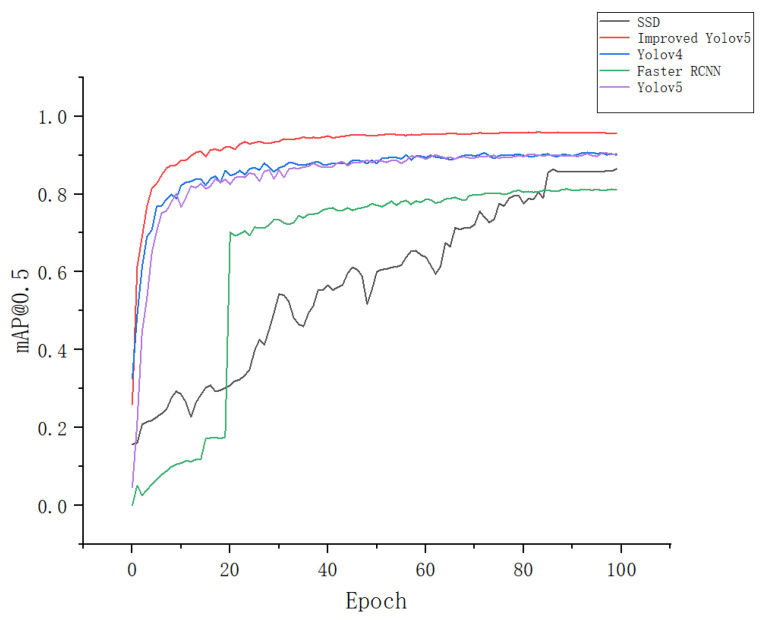
Accuracy variation of five object detectors.

**Figure 12 plants-12-00169-f012:**
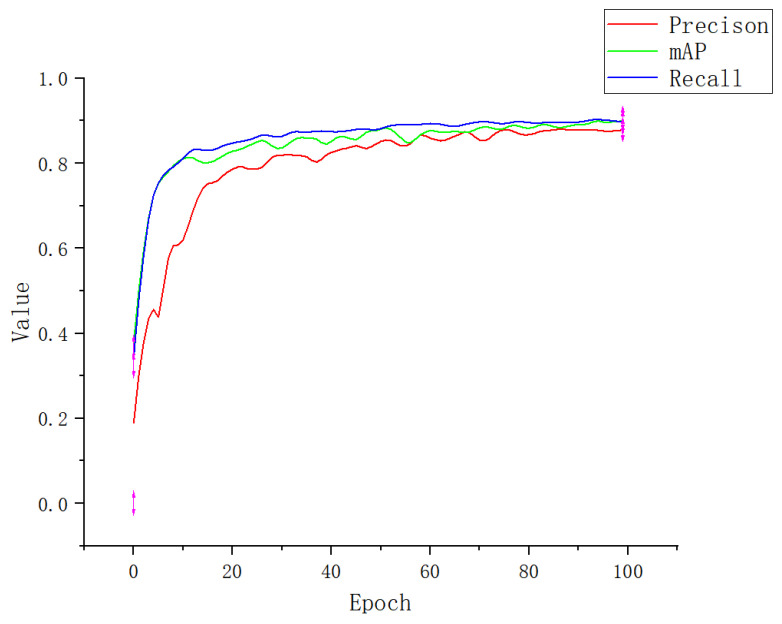
Performance index curve of SE.

**Figure 13 plants-12-00169-f013:**
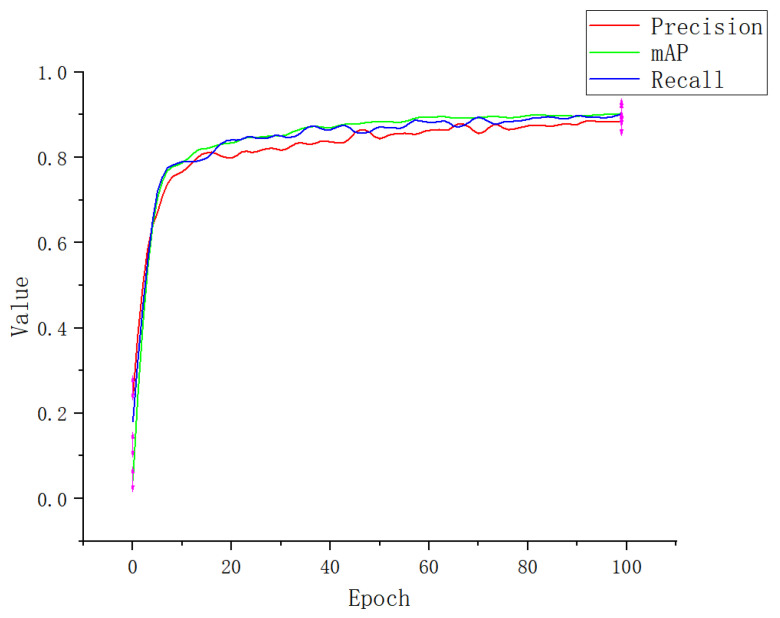
Performance index curve of ECA.

**Figure 14 plants-12-00169-f014:**
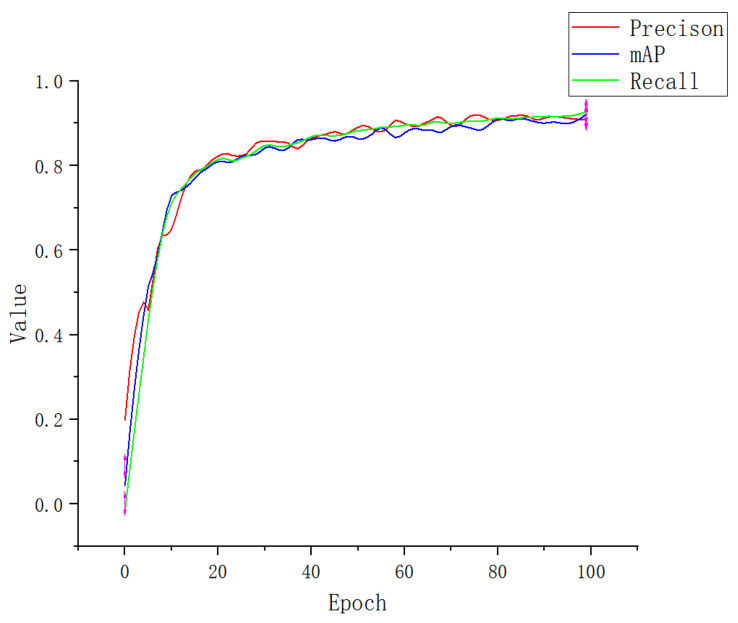
Performance index curve of CBMA.

**Figure 15 plants-12-00169-f015:**
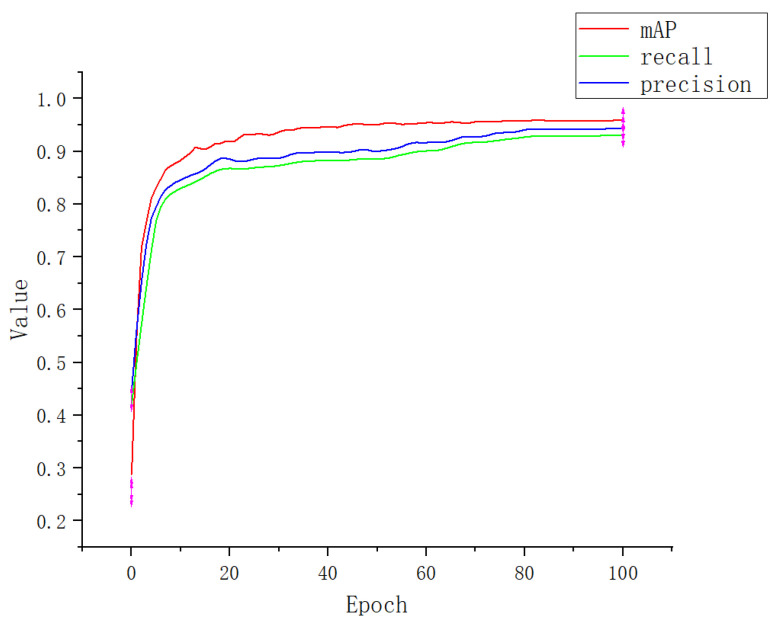
Performance index curve of CA.

**Figure 16 plants-12-00169-f016:**
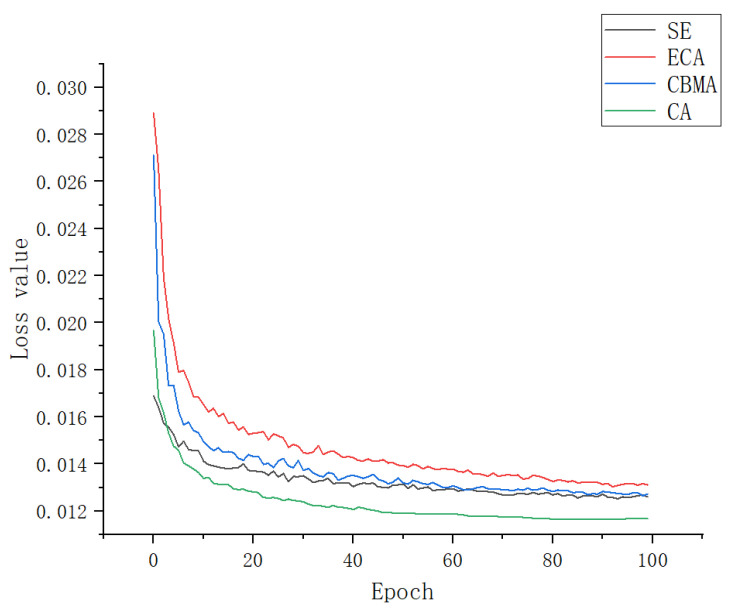
Loss value of each module.

**Figure 17 plants-12-00169-f017:**
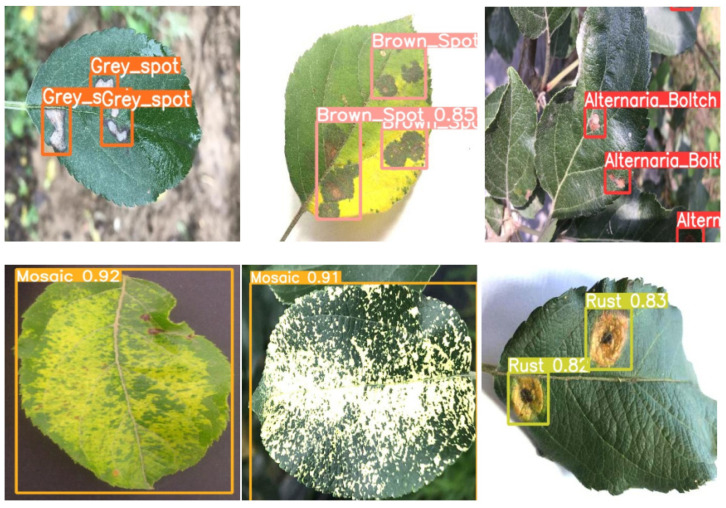
Apple leaf disease detection images.

**Table 1 plants-12-00169-t001:** Details of the apple leaf diseases dataset.

Classes	Number of Training Data	Number of Test Data	Label Number
Alternaria blotch	2000	500	0
Brown spot	2000	500	1
Gray spot	2000	500	2
Mosaic	2000	500	3
Rust	2000	500	4

**Table 2 plants-12-00169-t002:** Performance comparisons of different models.

Model	Precision/%	Recall/%	mAP@0.5/%
SSD	86.2	88.7	86.5
Faster RCNN	82.1	84.9	81.2
YOLOv4	84.5	86.7	90.3
YOLOv5	87.6	90.3	89.8
Apple-Net	93.1	94.4	95.9

**Table 3 plants-12-00169-t003:** Performance comparison of different attention modules.

Module	Precision/%	Recall/%	mAP/%
YOLOv5s + SE	87.9	90.4	90.1
YOLOv5s + ECA	91.2	91.6	92.8
YOLOv5s + CBAM	89.5	90.9	90.3
YOLOv5s + CA	93.1	94.4	95.9

**Table 4 plants-12-00169-t004:** Comparison with existing methods for apple disease detection.

References	Methods/Models	Categories	Images	mAP@0.5	Accuracy
[6]	SVM	4	899	/	92.49
[14]	MEAN-SSD	5	26,767	83.12	97.07
[43]	VGG-Net	4	2446	/	99.01
[44]	Res-Net	4	4174	/	83.75
[51]	KNN	2	744	/	96.40
[52]	DenseNet-201	4	2537	/	98.75
[53]	DP-CNNS	5	26,377	78.8	97.14
[54]	MGA-YOLO	4	8838	89.3	94.80
[55]	MSO Res-Net	5	11,397	89.6	95.70
Proposed	Apple-Net	5	15,000	95.9	96.74

## Data Availability

The apple leaf dataset is publicly available online at https://aistudio.baidu.com/aistudio/datasetdetail/11591 (accessed on 26 December 2022).

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
