# Peer review of "Apple-Net: A Model Based on Improved YOLOv5 to Detect the Apple Leaf Diseases"

_plants, 2022, doi:10.3390/plants12010169_

Round 1
Reviewer 1 Report (Previous Reviewer 1)
The revised paper is improved and is acceptable for publication
Author Response
Thank you very much for taking time out of your busy schedule to review our paper.

Reviewer 2 Report (Previous Reviewer 2)
Dear Authors, thank you very much for a new opportunity to review your manuscript "Apple-Net: a model based on improved yolov5 to detect the apple leaf diseases" (plants-2108884) demonstrated a potential classification and identification of diseases in apple leaves. In addition, the authors optimized the “yolov5” models to improve the final detection.
The author improved all manuscript and I happy with these alterations. Introduction, material and methods, the results and discussion in single topic and conclusion are good. However, two main points even need attention.
#1. English need minor corrections in phrases and syntax. Maybe an English Native.
#2. Please. Check all manuscript for standardization of nomenclature, dots, legends (tables and figures).
Best regards
Author Response
Thank you very much for taking time out of your busy schedule to review our article and put forward some meaningful suggestions. We have made detailed improvements to these suggestions in the response PDF document.

Reviewer 3 Report (Previous Reviewer 3)
This study focused on the improved yolov5 to detect the apple leaf diseases. The author did not accepted my previous suggestions so I suggest rejecting this manuscript, due to the severely flawed structure, inproper language for the scientific paper in general and indefinite expressions that should not be part of any scientific paper such: „More and more algorithms”, “accuracy in data training set and test set.”, “faster detection speed”, “they have achieved good results in accuracy” etc. What that means?
The Discussion section is now part of Results sections, but the aim of Discussion section is to obtain the contribution of the study and compare it to other scintific paper, not to explain graphs. The Discussion section should objectively demonstrate the scientific importance of work. The lack of Discussion severely impairs the scientific contribution of this study.
In the whole paper, there are too few citations, especially in the Materials and Methods section when explaining formulas. This implies that the authors developed the majority of the core methodology themselves, which is very unlikely.
In the title of the paper is written Apple-Net model as well as the keyword, but is nowhere else mentioned. Where is it in the Results section?
The network structure diagram of the network is too small, and not visible as well as other images in this paper, such as Figure 16. which should be the most important part of the study.
„In addition, deep learning can combine feature extraction and classification, reduce the training cost of machine learning method” Is that really true???
In Conclusion section, there is such a mess:
“In our view, the model can also be applied to other agricultural production fields such as disease detection of oranges, pears and other fruits. It can also be used for disease detection of wheat, rice and other crops. In the future, we can further improve our model, improve other structures to make our model lighter, reduce training parameters, and speed up model training. The model can be applied to different application scenarios.”
Three times is stated obvious thing, but in different way.
Author Response
Thank you very much for reviewing our paper and making meaningful suggestions in your busy schedule. We have revised our paper according to these suggestions, and the specific contents are shown in the PDF document.

Reviewer 4 Report (New Reviewer)
-The paper should be interesting ;;;
-it is a good idea to add a block diagram of the proposed research/review (step by step);;;
-it is a good idea to add more photos of measurements, sensors + arrows/labels what is what (if any);;;
-What is the result of the analysis?;;
-figures should have high quality. ;;;;;
-references in the text to figures should be added;;;
-text should be formatted;;;;
-please add photos of the application of the proposed research, 2-3 photos ;;;
-what will society have from the paper?;;
-labels of figures should be bigger;;;;
-Is there a possibility to use the proposed research for other topics;;;
-please compare the advantages/disadvantages of other approaches;;;;;;
-references should be from the web of science 2020-2022 (50% of all references, 30 references at least);;;
-Conclusion: point out what have you done;;;;
-please add some sentences about future work;;;
Author Response
Thank you very much for reviewing our paper and making meaningful suggestions in your busy schedule. We have revised our paper according to these suggestions, and the specific contents are shown in the PDF document

Round 2
Reviewer 3 Report (Previous Reviewer 3)
I suggest sending this manuscript for an official English language check due to such an inexact and unformal language for the scientific paper, causing very poor English overall. This paper needs completely refactorization of almost each of sentence because of improper language. I write some specific comments just for Abstract and the first part of the Introduction as some examples of poorly written paper.
Also, some inaccurate information is written like:
“In recent years, the field of artificial intelligence in agriculture is developing slowly.”
without any reference.
It is mandatory for authors to study scientific literature before writing scientific papers and to familiarize themselves with structure and purpose of scientific papers.
I repeat that the aim of Discussion section is to obtain the contribution of the study and compare it to other scientific paper. It is too vague and inadequate whose lack of severely impairs the scientific contribution of this study.
Specific comments:
line 11-12: I would suggest to remove this sentence. In literature, there are some very accurate deep learning algorithms both for classification and detection of diseases.
line 20-21: I would suggest to remove this sentence, “good results” are relative term.
line 23-24: Instead of “well solve”, the better term is more optimized or more efficient than…
line 28: The same words are repeated, “worldwide” and “in the world”. The sentence needs reference.
line 29: What means “the most productive fruit”?
line 31: Will the leaves always be attacked by pest and diseases? Reference needed.
line 32-33: “Once apple leaves are infected, the host will increase, and then the whole
tree will be infected in a large area.” What a poor English.
line 35: Why is “Moreover” written with capital letter?
line 70-71: “In the process of manual diagnosis of diseases, the fruit growers will make false judgments and missed judgments due to fatigue…” Farmers are tired?
Author Response
Thank you for your valuable comments, and we have revised this.

Reviewer 4 Report (New Reviewer)
figures should have high quality;;;
block diagram of research should be added, step by step;;
Author Response
Thank you for your valuable comments, and we have revised this.

Round 3
Reviewer 3 Report (Previous Reviewer 3)
The authors performed the most of my previous suggestions. However, the Discussion is still suboptimal and should be additionally expanded and improved.
Author Response
Thank you for your valuable comments. We have made improvements. See PDF for details

This manuscript is a resubmission of an earlier submission. The following is a list of the peer review reports and author responses from that submission.
Round 1
Reviewer 1 Report
Dear Authors
In your paper you present an improved version of yolov5 with a better mean average precision and a better precision value. Based on the statements in the introduction I expected also a faster detection of the apple leaf disease, but in the conclusion you admitted that the procedure takes a long time and an improved version is required. Try to spend some time on improvement proposal.
I find your paper hard to read, so I recommend to improve the language. I propose to modify the paper style, to become readable for the readers who are less familiar with the topic and try to expand the paper with the explanations and reduce it with less important details which are boring and hard to understand for an average reader of the magazine.
Nevertheless the reported results are encouraging.
I wish you good luck.
Reviewer 2 Report
The manuscript "Apple-Net: a model based on improved YOLOv5 to detect apple leaf diseases" (plants 2069129) demonstrated a potential classification and identification of diseases in apple leaves. In addition, the authors optimized the “yolov5” models to improve the final detection. I think topic its interesting, but not with these "displayed".
The introduction, material and methods its need correction and improved English language corrections. However, results its very confuse and there is no discussion topic. Many references its not correct following “Author Instruction” in Plants.
Some points:
#1: There is a scope for improvement in the introduction section: a) additional emphasis on the significance of the study, b) scientific and economic contribution of the paper; c) prospectively to other plants to agronomic interest. Maybe a one paragraphs with potential economic by important this manuscript to science crop;
#2. Your hypothesis is not clear in last paragraph to introduction. Pease write “Our hypothesis was…” and What is your objective with this study?
#3: Please. All standardization of nomenclature equipment/software when necessary. Example: Fabricant, City, State, Country (three-letter). Check all manuscript.
#4. As the work is very interesting, various statistical and graphical tools are used therefore;
#5. Alphabetic order keywords;
#6. Please, check all references. Crosscheck all the references.
#7. What is more significance of your manuscript?
#8. Please, check standardizing to scientific and notation terms;
#9. Author Contributions, Please, check following system submission;
*Maybe an abbreviation list, should be improved and clear our understand.
Minor point: spaces between references [xx] and phrases. Many sentences, its not comprehensive.
L.91-93. Why is it important? What it influences by algorithm testing?
L213. What’s this “CSPDarknet53”?
What is the difference between fast and speed detection?
Figure 7. Its minor quality;
Why many formulae, if not shown a discussion or influence in your results?
Why “The structure of this part is shown in Figure 10.”. What do you mean by this scheme?
What is its benefit to SVM, CNN, FPN, ECA, and other algorithms?
What is sample number? 1 plant /1 hectare of the plants?
I suggestion author check all manuscript and rewrite/reformulation, because, the topic its interesting but its manuscript doesn’t its. The manuscript its very confused and not demonstrated a based scientific criterion to publication in currently.
Best regards
Reviewer 3 Report
This study focused on the improved yolov5 to detect the apple leaf diseases. I suggest rejecting this poorly written manuscript, due to the severely flawed structure and inproper language for the scientific paper in general.
Firstly, the structure of the paper is wrong. The Results section comes before Materials and Methods. Moreover, the Discussion section is not written anywhere, nor as a part of the Results section. The lack of Discussion severely impairs the scientific contribution of this study.
Secondly, in the whole paper, there are too few citations, especially in the Materials and Methods section. This implies that the authors developed the majority of the core methodology themselves, which is very unlikely.
Thirdly, in the title of the paper is written Apple-Net model as well as the keyword, but is nowhere else mentioned. Where is it in the Results section?
The network structure diagram of the network is too small, and not visible as well as other images in this paper. In the paper, there is only one curve in the training loss graph, but there are far more networks in the table. In the Introduction, there are only stated some „similar“ papers without any other comment, and is too short. What do you want to achieve in this paper? Where is any citation regarding used formulas in Materials and Methods? What is meant by „Training Number“ and „Test Number“? The dataset is not balanced between classes.
As the paper is too poorly written, I will comment just one line for specific comments:
line 135: „As we all know…” This is a scientific paper, and this way of writing is not appropriate. This sentence is written in other sections of the paper as well.